# Fine Particle Adsorption Capacity of Volcanic Soil from Southern Kyushu, Japan

**DOI:** 10.3390/nano13030568

**Published:** 2023-01-30

**Authors:** Naoaki Misawa, Kentaro Yasui, Kentaro Sakai, Taichi Kobayashi, Hideki Nagahama, Tomohiro Haraguchi, Satomi Sasaki, Vetchapitak Torrung, Taradon Luangtongkum, Takako Taniguchi, Kentaro Yamada, Makoto Minamimagari, Toshihiro Usami, Hiroyuki Kinoshita

**Affiliations:** 1Center for Animal Disease Control, University of Miyazaki, 1-1 Gakuen-Kibanadai-Nishi, Miyazaki 889-2192, Japan; 2National Institute of Technology, Kagoshima College, 1460-1 Shinko, Hayato-cho, Kirishima 899-5193, Japan; 3Organization for Promotion of Research and Industry-Academic Regional Collaboration, University of Miyazaki, 1-1 Gakuen-Kibanadai-Nishi, Miyazaki 889-2192, Japan; 4Department of Engineering, University of Miyazaki, 1-1 Gakuen-Kibanadai-Nishi, Miyazaki 889-2192, Japan; 5Department of Veterinary Public Health, Chulalongkorn University, Phayathai Road, Pathumwan, Bangkok 10330, Thailand; 6Department of Veterinary Science, University of Miyazaki, 1-1 Gakuen-Kibanadai-Nishi, Miyazaki 889-2192, Japan; 7Nanken Kogyo Co., Ltd., 5629-2 Yamada-Karuishi, Yamada-cho, Miyakonojo 889-4601, Japan; 8Graduate School of Engineering, University of Miyazaki, 1-1 Gakuen-Kibanadai-Nishi, Miyazaki 889-2192, Japan

**Keywords:** volcanic soil, dye, NO_2_, SO_2_, phosphoric acid, *Escherichia coli*, adsorption tests

## Abstract

“Akahoya” is a volcanic soil classified as a special soil deposited in Kyushu, Japan. Many of its properties are not yet clearly understood. We found that Akahoya had the potential to adsorb bacteria in cattle feces, which prompted us to investigate its material properties and perform experiments to comprehensively evaluate its adsorption performance for various fine particles such as acidic and basic dyes, NO_x_/SO_x_ gas, and phosphoric acid ions, in addition to bacteria. Akahoya had a very high specific surface area owing to the large number of nanometer-sized pores in its structure; it exhibited a high adsorption capacity for both NO_2_ and SO_2_. Regarding the zeta potential of Akahoya, the point of zero charge was approximately pH 7.0. The surface potential had a significant effect on the adsorption of acidic and basic dyes. Akahoya had a very high cation exchange capacity when the sample surface was negatively charged and a high anion exchange capacity when the sample surface was positively charged. Akahoya also exhibited a relatively high adsorption capacity for phosphoric acid because of its high level of Al_2_O_3_, and the immersion liquid had a very high Al ion concentration. Finally, filtration tests were performed on *Escherichia coli* suspension using a column filled with Akahoya or another volcanic soil sample. The results confirmed that the *Escherichia coli* adhered on the Akahoya sample. The results of the *Escherichia coli* release test, after the filtration test, suggested that this adhesion to Akahoya could be phosphorus-mediated.

## 1. Introduction

The Japanese forestry industry has declined because it has been unable to recover the costs of thinning and other nursery work, logging, and removal of timber. To address these problems, Miyazaki Prefecture in Kyushu introduced agroforestry [1], which involved expanding cattle grazing areas to include not only pastures but also planted forests. However, concerns have been raised that cattle grazing in planted forests would lead to their feces contaminating streams and ultimately drinking water.

The waters surrounding an artificial forest where cattle were grazing were surveyed, and no bacterial contamination by cattle feces was identified. This led to the working hypothesis that the bacteria in cattle feces may be absorbed by soils in the grazing areas. Volcanic soils were deposited underground in this area. In particular, the soils contained white or light brown “Akahoya,” which does not retain water and adsorbs phosphate ions essential for plant growth [2,3,4]. One possible candidate soil for adsorbing bacteria in cattle feces is this Akahoya. 

Akahoya is volcanic ash that originated from the great eruption of the Kikai Caldera about 7300 years ago [5]. It was ejected at the same time as the Koya pyroclastic flow, and it then fell to the ground. Research into strata known to contain Akahoya around Mt. Kirishima in Miyazaki Prefecture showed that similar strata were distributed throughout Japan, including Imogo near Hitoyoshi City in Kumamoto Prefecture, Akabokko in Tanegashima Island in Kagoshima Prefecture, and Onji in southern Shikoku [4]. A further study confirmed that these soils were volcanic ash ejected from the Kikai Caldera. All the soils in Japan are classified as special soils because they are prone to landslides during heavy rains and are unsuitable for agriculture [3]. 

The special volcanic soils are rich in active aluminum (Al) and iron (Fe) [6,7]. Active Al is highly reactive with phosphate ions [8,9,10,11,12] and exists mainly as allophane, imogolite, and aluminum–humus complexes [13,14]. Imogo contains quasi-crystalline imogolite [15,16,17,18,19,20,21,22], which is similar to allophane, as the main mineral, and it is characterized by a composition of alumina and silica with high levels of activated aluminum (Al) as well as a high specific surface area. 

Other volcanic soils such as Akadama and Kanuma in Kanto, Japan also exhibit phosphoric acid adsorption [23,24,25,26]. However, these soils are not classified as special soils and have allophane as the main mineral. In other words, Imogo is not the same kind of soil as Akadama and Kanuma. In addition, Akahoya and Imogo do not seem exactly the same soil either. Akahoya seems to have unique properties not found in other volcanic soils [4], which may be attributed to diversity in the depositional patterns of volcanic ash. After deposition, the weathering products of volcanic ash change depending on the environmental conditions such as the climate, topography, and organisms, which result in soils that vary among regions. The properties of global volcanic ash soils also differ among each other depending on the location [26,27,28,29,30,31,32,33].

At present, many Akahoya properties are not clearly understood. There is also no report on whether volcanic soil has the ability to adsorb *E. coli*. In general, the soils containing allophane or imogolite as the main mineral have shown an excellent ion exchange performance [34,35,36,37] as well as phosphate adsorption. The soils might also have a high adsorption capacity for various harmful gases and fine particles because of their high specific surface areas. In addition, if it is found that Akahoya has the ability to adsorb *E. coli*, the soil could be used as a material for controlling *E. coli* outflow from livestock farms or as a material for water purification. The use of Akahoya as an absorbent will greatly contribute in maintaining a clean living environment. 

Therefore, in this study, to investigate whether Akahoya has noteworthy properties that can make it useful as an absorbent in agricultural and industrial fields, we first performed experiments to comprehensively evaluate the chemical and physical properties and cation/anion exchange, gas adsorption, and phosphoric acid adsorption performances of Akahoya. Next, we performed tests to verify the adhesion of *E. coli* to Akahoya and to investigate a factor of this adhesion.

## 2. Materials and Methods

### 2.1. Sample Production

In powder form, Akahoya is unsuitable as an adsorbent of fine particles or dyes. Therefore, we fired powdered Akahoya to produce granular ceramic samples with predetermined shapes. Figure 1 shows the fabrication process of the ceramic samples used in various adsorption tests:Akahoya was crushed by using a rotary mill and was then sifted through a 0.3 mm mesh screen.The crushed soil was pressed into a mold at 10 MPa to form solids with a diameter of 74 mm and thickness of 50–60 mm.The molded samples were heated at a firing temperature of 923–1373 K in an electric furnace (KY-4N, Kyoei Electric Kilns Co., Ltd., Tajimi, Japan).After firing, the ceramic samples were crushed, and particles with sizes of 0.5–1.0 mm or 1.4–2.0 mm were selected.

**Figure 1 nanomaterials-13-00568-f001:**
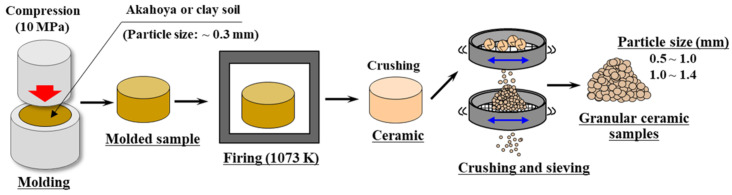
Fabrication of Akahoya ceramic samples for various adsorption tests.

### 2.2. Measurement of Material Properties

#### 2.2.1. Chemical Composition and Crystal Structure

To clarify the material properties of Akahoya, we measured the chemical compositions of several other kinds of volcanic soils as well: Shirasu [38,39,40,41], Bora, and a clay possessing a chlorite group as the main minerals. These soils were mined from the area surrounding Mt. Kirishima in southern Kyushu, Japan. Figure 2 shows microscopy images of each soil. Akahoya (Nanken Kougyo Co., Miyakonojo, Japan) is a reddish soil. Shirasu (Nanken Kougyo Co., Miyakonojo, Japan) comprises glassy powder and fine, very hard stones. Bora (Nanken Kougyo Co., Miyakonojo, Japan) comprises granular pumice with some degree of hardness. The clay (Sougoo Co., Miyakonojo, Japan) is typically used as a raw material for bricks and tiles. We measured the chemical compositions of the samples by using an energy-dispersive X-ray analyzer (EDX-720, Shimadzu Corporation, Kyoto, Japan). Then, we used X-ray diffraction (XRD) to examine the crystal structures of the soils by using an X-ray crystal structure analyzer (X’Pert-Pro MRD, Malvern Panalytical, Enigma, UK).

#### 2.2.2. Allophane/Imogolite Quantities

We determined the amount of allophane in Akahoya by the alternate dissolution method using hydrochloric acid (HCl, Nakarai Tesque Corporation, Kyoto, Japan) and sodium hydroxide (NaOH, Nakarai Tesque Corporation, Kyoto, Japan) of Kitagawa [42,43] with minor modification. This method is based on the principle that amorphous materials such as allophane and imogolite are rapidly reduced in weight by alternating dissolution with acid and alkali compared to crystalline clay minerals. After an Akahoya sample was dried at 453 K for 40 min, 80 mg of the sample was weighed and was placed in a 30 mL glass screw-top centrifuge tube with a body diameter of 28 mm (AS ONE, Tokyo, Japan). An 8 mL aliquot of 8 M HCl was added, and the tube was shaken at 180 rpm for 30 min at 293 K. The tube was then centrifuged at 200 g for 5 min, and the supernatant was discarded. The precipitate was washed twice with 8 mL of distilled water by centrifugation at 200 g for 5 min. Next, an 8 mL aliquot of 0.5 M NaOH was added to the sediment, and the tube was immersed in a hot-water bath at 453 K for 5 min. Then, the tube was centrifuged at 200 g for 5 min, and the supernatant was discarded. The tube was then washed twice by using distilled water as described above. To dry the Akahoya samples, the tube was heated at 453 K for 2 h. The glass tube and soil were then weighed to calculate the rate of weight loss. This procedure was repeated five times. For the control samples, sterilized distilled water was used instead of HCl and NaOH. Each experiment was performed in triplicate.

#### 2.2.3. pH and Al and Fe Ion Concentrations

The pH of the immersion liquid may affect the dye adsorption of a sample. In addition, allophane is known to be rich in activated Al and activated Fe, whose ions react easily with phosphoric acid. Therefore, to investigate the factors affecting the dye adsorption and phosphorus adsorption of Akahoya, we first measured the pH of the immersion liquid. We then measured the Al and Fe ion concentrations in the immersion liquid.

First, 5 g of an Akahoya sample with a particle size of 0.5–1.0 mm and that was fired at 973 K was immersed in 50 mL of distilled water. The pH was measured by using a pH meter (HM-25R, DKK-TOA Corporation, Tokyo, Japan). The Al and Fe ion concentrations of the immersion liquid were measured by using an inductively coupled plasma (ICP) emission spectrometer (ICPS-8100, Shimadzu Corporation, Kyoto, Japan). Then, 8 g of an Akahoya sample with a particle size of 0.5–1.0 mm that was fired at 1073 K was immersed in 50 mL of distilled water. This was stirred by using a stirring device at a speed of 300 rpm for 30 min, and the Al and Fe ion concentrations in the immersion liquid were measured. The sample mass was determined according to the procedure used for the phosphoric acid test (see Section 2.3.3). The Al and Fe ion concentrations in the immersion liquid of the clay sample fired at 1073 K were measured in the same way, and the ion concentrations of both samples were compared.

#### 2.2.4. Apparent Porosity, Specific Surface Area, Pore Size Distribution, and Compressive Strength

The specific surface area and porosity of the Akahoya ceramic samples decreased with the firing temperature because of the sintering process. An appropriate firing temperature needs to be selected to maximize the specific surface area and porosity. Therefore, we investigated the relationships between the firing temperature of the sample and the specific surface area, porosity, and compressive strength. Samples with a diameter of 30 mm and length of about 60 mm were fired at temperatures of 973–1423 K. The specific surface area and pore size distribution of the samples were measured by using a high-precision gas/vapor adsorption measurement instrument (BELSORP-max, MicrotracBEL Corp., Osaka, Japan). The apparent porosity was measured according to JIS R 2205-1992. Compressive tests were performed by using a universal testing machine (AG-X50kN, Shimadzu Corp., Kyoto, Japan) with a crosshead speed of 0.5 mm min^−1^. The compressive strength was obtained by dividing the measured maximum compressive load by the cross-sectional area of the samples.

#### 2.2.5. Zeta Potential

The surface potential of a solid greatly affects the cation or anion exchange performance in a liquid. Therefore, we measured the zeta potential of Akahoya samples by using a surface potential measurement device (Zetasizer Nano ZS, Malvern Panalytical Ltd., Worcestershire, UK).

### 2.3. Adsorption Tests

#### 2.3.1. Dye Adsorption

To investigate the ion exchange performance, we performed a dye adsorption test as shown in Figure 3 [44,45]. Three dyes were selected to represent basic, acidic, and direct (azo) dyes methylene blue (MB: C_16_H_18_N_3_SCl) (FUJIFILM Wako Pure Chemical Corporation, Osaka, Japan), Orange II (HOC_10_H_6_N:NC_6_H_4_SO_3_Na) (FUJIFILM Wako Pure Chemical Corporation, Osaka, Japan), and Congo red (C_32_H_22_N_6_Na_2_O_6_S_2_) (FUJIFILM Wako Pure Chemical Corporation, Osaka, Japan), respectively [46,47,48]. Akahoya and clay samples fired at 1073 K and unfired Bora and Shirasu samples were used. Each sample had a particle size of 0.5–1.0 mm. The reduction rates of the dye concentration for the samples were compared. The test procedure was as follows:Samples with a mass of 5 g (i.e., the same mass as the samples immersed in distilled water to determine the pH) were washed with distilled water and were dried in an electric furnace at 373 K for over 24 h.MB, Orange II, and Congo red powders were dissolved in distilled water to yield aqueous solutions each with a concentration of 1 × 10^−4^ mol/L.Then, 1 g of the granular samples was placed in a beaker containing 50 mL of the aqueous solution, and the aqueous solution was stirred by using a stirring device (EYLA ZZ-1010, Rikakikai Co., Ltd., Tokyo, Japan) at a speed of 150 rpm.The dye concentration and pH value of the aqueous solutions were measured after 1, 10, 30, 60, and 120 min.The dye concentration in the aqueous solution was measured by using a drainage analyzer (NDR-2000, Nippon Denshoku Industries Co., Ltd., Tokyo, Japan). The absorbance of the aqueous solution was determined. Then, the absorbance was used with a calibration curve to calculate the corresponding dye concentration. The pH of the aqueous solution was measured by using a pH meter (HM-25R, DKK-TOA Corporation, Tokyo, Japan).

**Figure 3 nanomaterials-13-00568-f003:**
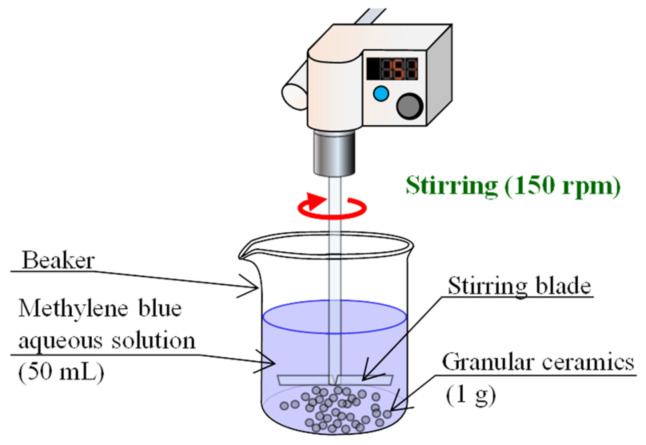
Schematic diagram of the dye adsorption test.

#### 2.3.2. NO_2_ and SO_2_ Adsorption

NO_2_ and SO_2_ are the main combustion gases of fossil fuels. We tested whether Akahoya can adsorb these gases, as shown in Figure 4 [49]. The experimental apparatus comprised a NO_2_ or SO_2_ gas cylinder, air cylinder (Standard gas, Hinode-sanso, Nobeoka, Japan), 50 L chemical resistance sampling bag (Tedlar Tech-Jam, Osaka, Japan), air pump (APN-085E-D2-W, Iwaki, Tokyo, Japan), digital flowmeter (CMS0020BSRN, Azbil, Tokyo, Japan), test tube, and instrument for measuring the NO_2_ or SO_2_ concentration (XPS-7, New Cosmos Electric, Osaka, Japan). Akahoya and clay samples fired at 1073 K and the unfired Bora sample were used in the gas adsorption tests. Each sample had a particle size of 1.0–1.4 mm. The gas-concentration reduction rates of the samples were compared. The test procedure was as follows:Samples were washed by using distilled water and were dried in an electric furnace at 378 K for over 24 h before the gas adsorption tests.For the NO_2_ adsorption test, the 50 L sampling bag was filled with 20 L of gas at a concentration of approximately 6 ppm.For the SO_2_ adsorption test, the 50 L sampling bag was filled with 10 L of SO_2_ gas at a concentration of approximately 10 ppm and 10 L of standard air gas. To homogenize the concentration, the mixture gas was circulated at a flow rate of 2 L/min for 20 min.A 5 g soil sample was placed in a test tube. We allowed the NO_2_ or SO_2_ gas to pass through the test tube containing the sample at a flow rate of 1.0 L/min and circulated the gas in the circuit for up to 4 h.The NO_2_ or SO_2_ concentration in the sampling bag was measured at intervals of 30 min. The pump was momentarily stopped during each measurement.

**Figure 4 nanomaterials-13-00568-f004:**
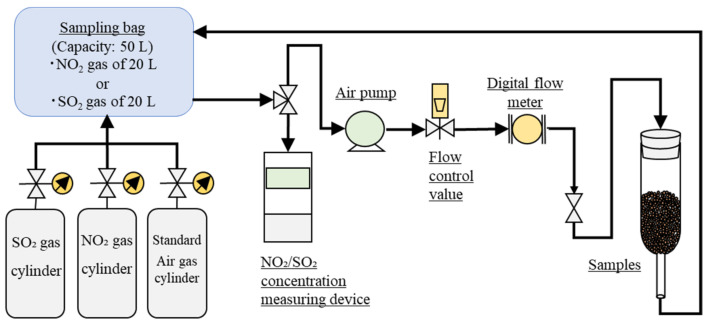
Schematic diagram of the NO_2_ and SO_2_ adsorption tests.

#### 2.3.3. Phosphoric Acid Adsorption

The phosphoric acid adsorption test was conducted as follows. First, 5, 16, 20, or 24 g of a soil sample was placed in a beaker containing 100 mL of potassium dihydrogen phosphate (KH_2_PO_4_, Nakarai Tesque Corporation, Kyoto, Japan) aqueous solution at a concentration of 200 ppm. The beaker was stirred with a stirring device at a speed of 300 rpm. Then, the phosphorus concentration of the aqueous solution after 5, 10, and 30 min was measured by using an ICP emission spectrometer (ICPS-8100, Shimadzu Corporation, Kyoto, Japan). The Akahoya samples were fired at 973, 1073, or 1173 K, and the clay sample was fired at 1073 K. The samples had a particle size of 1.0–1.4 mm. The reduction rates in the phosphorus concentration with the Akahoya and clay samples were compared.

#### 2.3.4. Filtration Test of *E. coli* Suspension

Akahoya was sieved to collect particle sizes of 1 mm or less and was sterilized by dry heat at 453 K for 30 min. A column with a diameter of 15 mm and height of 100 mm (Econo-Pac Disposable Chromatography Columns, BioRad, Tokyo, Japan) was filled with 5 g of a soil sample. Then, 10 mL of an *E. coli* (strain ATCC 25922) suspension in 10 mM phosphate-buffered saline (pH 7.2) (Nakarai Tesque Corporation, Kyoto, Japan) with a concentration of 3.0 × 10^7^ cfu/mL was added from the top of the column. The liquid that passed through the column was collected for enumeration of *E. coli* according to the direct plating method using DHL selective agar plates (Nissui Co., Tokyo, Japan). The same operation was repeated five times for one experiment. Shirasu (i.e., plagioclase quartz and coarse-grained volcanic sediment) from southern Kyushu, Japan was used as a control. Each experiment was performed in triplicate.

#### 2.3.5. Release Test

To verify whether *E. coli* adheres to Akahoya due to phosphorus mediation, a release test was performed. First, 10 g of Akahoya was packed in the column as described above, and 1 × 10^6^ cfu of *E. coli* was adhered onto the soil. The column was inoculated with 10 mL of 1 M phosphate buffer (pH 7.2) from the top of the column nine times, and each passage was collected. Then, the number of *E. coli* in each pass through the suspension was enumerated quantitatively by using DHL agar plates. This culture method had a detection limit of 10 cfu. The same test was conducted by using distilled water instead of the phosphate buffer as a control. Each experiment was performed in triplicate.

## 3. Results and Discussion

### 3.1. Chemical and Physical Properties

#### 3.1.1. Chemical Composition, pH, Al and Fe ion Concentrations, and Crystal Structure

Table 1 presents the chemical compositions of the special volcanic soils and clay after firing at 973 K. For all soils, the main components were silica and alumina. Shirasu, Bora, and the clay had similar compositions. However, Akahoya had lower SiO_2_ and higher Al_2_O_3_ and Fe_2_O_3_ contents than the other soils.

Figure 5 shows that the pH of the immersion liquid of Akahoya was neutral or weakly alkaline. The immersion liquid for the clay was also weakly alkaline, although not shown in the figure.

Table 2 presents the Al and Fe ion concentrations in 50 mL of immersion liquid containing 8 g of a sample with a particle size of 0.5–1.0 mm that was fired at 1073 K. The immersion liquid containing Akahoya had a much higher Al ion concentration than the immersion liquid containing clay. This result suggests that Akahoya has a high adsorption capacity of phosphoric acid.

We investigated the difference in mineral compositions of Akahoya, imogolite, and allophane. Imogolite is the primary clay mineral in Imogo, which also originates from the Kikai Caldera as with Akahoya. In imogolite, the molar ratio of SiO_2_ and Al_2_O_3_ is about 1:1, and the Fe content is extremely low [15]. The immersion liquid for imogolite had a weakly acidic pH. In contrast, the molar ratio of SiO_2_ and Al_2_O_3_ for allophane is about 1–2:1, and the immersion liquid has a weakly acidic pH. Table 1 indicates that the molar ratio of SiO_2_ and Al_2_O_3_ in Akahoya was about 2.7:1. This confirmed that Akahoya is not exactly the same as Imogo, because of the higher molar ratio of SiO_2_ and Al_2_O_3_. In addition, Akahoya contained a considerable amount of Fe. Although the mineral composition of Akahoya was similar to that of allophane, we also confirmed that Akahoya slightly differs from soils such as Akadama that mainly contain allophane because the pH of the immersion liquid was neutral or weakly alkaline.

#### 3.1.2. Allophane Quantity

Figure 6 shows the weight loss rate of Akahoya by the acid–alkali alternate dissolution method. The weight loss rate increased with the number of dissolution treatments and was 52.5% after the fifth treatment. When distilled water was used instead of acid and alkali as a control, the weight loss rate of Akahoya was less than 20% even after five treatments. This result confirmed that Akahoya contained 50% or more allophane. Note that the amount of allophane may also include imogolite and other amorphous minerals because this method quantifies clay minerals with low crystallinity [42].

#### 3.1.3. Crystal Structure and Surface Microstructures

Figure 7 shows the XRD profiles of clay and Akahoya. The unfired clay mainly contained quartz (SiO_2_) and the chlorite group as crystallized minerals, and the clay fired at 1073 K only contained quartz. Akahoya contained quartz and albite (NaAlSi_3_O_8_) as crystallized minerals. Recent research [23,50,51] has shown that allophane can be identified by XRD owing to small peaks near the diffraction angles of 26° and 40°. However, we were unable to identify these peaks in this study.

Figure 8 shows scanning electron microscopy (SEM) images taken by S-5500 (Hitachi High-Technologies Corporation, Tokyo, Japan) of the surface structures on the Akahoya sample. Platinum sputtering was performed to ensure appropriate conductivity on the sample surface. Akahoya mainly had a fibrous structure and very fine granular soils that looked like allophane. The fibrous structure is peculiar to imogolite and is considered a hollow tube [8]. These results indicate that Akahoya mainly contained imogolite. Wada also reported that imogolite and allophane usually coexist in volcanic soil [8,25]. Therefore, we concluded that Akahoya is a volcanic soil that mainly contains imogolite, allophane, and other amorphous minerals with a relatively high Fe content, and quartz and albite as crystalline minerals.

#### 3.1.4. Apparent Porosity, Specific Surface Area, Pore Size Distribution, and Compressive Strength

Figure 9a shows the specific surface areas of Akahoya, Shirasu, Bora, and clay. Unfired Akahoya possessed an extremely high specific surface area (144 m^2^/g) compared with the others. This is because Akahoya had a large number of nanometer-sized pores owing to the high contents of imogolite and allophane. The fired Akahoya samples also had a relatively high specific surface area up to a firing temperature of about 1023 K. Figure 9b shows the apparent porosities of the samples. Akahoya possessed an extremely high porosity of 50% or more up to a firing temperature of about 1323 K.

Figure 10a shows the pore size distributions of unfired Akahoya, Shirasu, Bora, and clay samples. Akahoya had more nanometer-sized pores. In contrast, Shirasu had few nanometer-sized pores. Bora had several tens of nanometer-sized pores. Clay had some nanometer-sized pores but much fewer than Akahoya. The high specific surface area of Akahoya can be attributed to the large number of nanometer-sized pores. Figure 10b shows the relationship between the pore size distribution and firing temperature of Akahoya. The nanometer-sized pores decreased with an increasing firing temperature, which, in turn, decreased the specific surface area. However, Akahoya possessed some nanometer-sized pores up to a firing temperature of 1173 K.

Figure 11 shows the compressive strengths of Akahoya samples that were fired at 973–1423 K. The average compressive strength was calculated from measurements of 10 or more samples. The error bars represent the standard deviation. Akahoya retained some compressive strength when fired at 973 K or higher. To obtain sintered products using Akahoya as a raw material, a firing temperature of 973 K or higher is needed. Therefore, we mainly used samples fired at 1073 K for the adsorption tests.

#### 3.1.5. Zeta Potential

Figure 12 shows the zeta potential of Akahoya samples that were unfired and fired at 1073–1373 K. The point of zero charge (PZC) of the unfired sample was approximately 7.0 pH. The PZC of the sample fired at 1073 K was 2.9 pH. The PZCs of the samples fired at 1273 or 1373 K were not measured in the pH range of 2.5–9.5. This result indicates that the surface of the unfired sample was positively charged when the aqueous solution was acidic and negatively charged when the solution was alkaline. The surface of the sample fired at 1073 K was negatively charged except when the aqueous solution was a strong acid. The surface of samples fired above 1273 K was negatively charged at all pH values. Thus, the surface potential of Akahoya has a significant effect on its adsorption of acidic and basic dyes. The above basic properties of Akahoya were used to conduct dye, NO_2_/SO_2_, phosphoric acid, and *E. coli* adsorption tests.

### 3.2. Dye Adsorption

#### 3.2.1. Results

Figure 13 shows the dye concentration reduction rates of the samples and the temporal change in pH of the dye solutions during the dye adsorption test. Akahoya had a significantly higher MB dye concentration reduction rate than clay. In contrast, Bora and Shirasu had very low reduction rates. The MB solution containing Akahoya was weakly alkaline, and the pH was almost constant over time. The MB solutions containing other samples were weakly acidic. Both Akahoya and clay did not adsorb Orange II at all. Akahoya did not adsorb Congo red at all, although the clay slightly adsorbed the dye. Both the Orange II and Congo red solutions containing Akahoya were weakly alkaline. These results indicated that Akahoya had a high adsorption capacity for MB, which is a basic dye. In contrast, it did not adsorb Orange II and Congo red, which are acidic and azo dyes, respectively.

#### 3.2.2. Ion Exchange Performance and Potential for Acidic Dye Adsorption by pH Adjustment

The above results showed that Akahoya only adsorbed MB very well among the dyes. Figure 12 indicates that the Akahoya sample fired at 1073 K had a negatively charged surface, and Figure 13 indicates that solutions containing Akahoya were weakly alkaline in all dye adsorption tests. Therefore, Akahoya adsorbed cations from the aqueous solution by ion exchange. MB existed as cations in the solution, which explains why it easily adsorbed onto Akahoya. In contrast, Orange II and Congo red existed as anions in their solutions, so they did not adsorb onto Akahoya. This also explains why clay only adsorbed MB well, because it had a cation exchangeable capacity of 5–40 [37,52]. Our previous study [45] showed that a material with a higher specific surface area has a more active ion exchange. Therefore, Akahoya had a higher MB concentration reduction rate than the clay because it possessed a very high porosity and extremely large specific surface area.

Figure 14a,b show the Orange II concentration reduction rates when the initial pH of the dye solution was adjusted to 2–6 by the addition of hydrochloric acid. Akahoya adsorbed Orange II well when the initial pH was adjusted to 2. This result appears to be because the positively charged surface of the sample adsorbed the acidic dye through anion exchange. Clay also slightly adsorbed the dye when the initial pH was adjusted to 2. The above results indicate that Akahoya also has a high adsorption capacity for acidic dyes when unfired or fired at a relatively low temperature.

Figure 14c shows the Congo red concentration reduction rate when the initial pH of the dye solution was adjusted to 4 or 6. The initial pH was not adjusted to 2, because the color of Congo red changes from red to dark blue when the pH is less than 3. Akahoya adsorbed a considerable amount of Congo red when the dye solution was adjusted to acidic. Therefore, Akahoya may also be applicable as an absorbent for azo dyes by pH adjustment. This would be useful because azo dyes are the most common type. However, the adsorption mechanism is clearly not ion exchange [53,54], because Congo red was adsorbed even though the surface of the Akahoya sample was negatively charged.

### 3.3. NO_2_ and SO_2_ Adsorption

Figure 15 shows the NO_2_ and SO_2_ concentration reduction rates of the samples. Each graph represents the average reduction rates calculated from measurements of three samples. The error bars represent the upper and lower limits of the measured values. The graph of “Without sample” shows the NO_2_ or SO_2_ concentration reduction rate when the sample was not placed in the test tube. The error in the NO_2_ or SO_2_ concentration reduction rate without the sample was attributed to the difference in humidity caused by the difference in air temperature. The concentration reduction rate of Akahoya was compared to those of clay and Bora, which is a volcanic pumice stone with a high NO_2_ adsorption ability [55]. 

Akahoya had a significantly higher NO_2_ concentration reduction rate than clay or Bora, which had roughly comparable NO_2_ concentration reduction rates. In contrast, Akahoya had a slightly higher SO_2_ concentration reduction rate than Bora and a significantly higher rate than clay. These results demonstrated that Akahoya had a high adsorption capacity for both NO_2_ and SO_2_. This may be because of the very high specific surface area and porosity of Akahoya. The adsorption mechanism may be electrical interaction between the NO_2_ and SO_2_ gases and the Akahoya surface because NO_2_ and SO_2_ are polar molecules.

### 3.4. Phosphoric Acid Adsorption

Figure 16 shows the changes in the phosphorus concentration and pH of the solutions containing Akahoya or clay fired at 1073 K. The results indicated that Akahoya adsorbed some phosphorus while clay adsorbed little. The pH of the aqueous solution containing Akahoya increased gradually over time and eventually changed from weakly acidic to almost neutral. This indicates that the aqueous solution was gradually neutralized by a chemical reaction.

We examined the influence of the firing temperature and sample amount of Akahoya on the phosphorus concentration. Figure 17a shows the change in phosphorus concentration of the aqueous solution with Akahoya samples fired at 973, 1073, and 1173 K. The samples adsorbed phosphorus when fired at temperatures below 1173 K. Figure 17b shows the change in the phosphorus concentration with the sample amount. For a 100 mL solution with a phosphorus concentration of 200 ppm, the phosphorus concentration was significantly reduced with 16 g of Akahoya or more.

### 3.5. Application of Akahoya to E. coli Removal

#### 3.5.1. *E. coli* Adherability

Figure 18 shows that no *E. coli* was recovered from when the bacterial suspension was passed through the column five times, and the inoculated *E. coli* was completely filtered by Akahoya. In contrast, Shirasu, which was used as the control substance, hardly filtered any *E. coli*, and the inoculum was recovered from the bacterial suspension passed through the column.

#### 3.5.2. Primary Influencing Factors

Figure 19 shows the number of *E. coli* (log cfu/ml) released from Akahoya after adding 1 M phosphate buffer or distilled water. The number of *E. coli* in the passage decreased as the number of phosphate buffer inoculations increased. On the other hand, when Akahoya that had adhered *E. coli* was inoculated with distilled water, no *E. coli* was found in any passage even after nine spike inoculations. The structure of the cell membrane of *E. coli* comprises phospholipids such as phosphoric acids and fatty acids [56]. These suggest that the adhesion of *E. coli* to Akahoya could be mediated by phosphorus.

### 3.6. Discussion

First, we discuss the ion exchange function of Akahoya. Akahoya had a high adsorption capacity for MB, which is a basic dye, when the sample surface was negatively charged. It also had high adsorption capacity for Orange II, which is an acidic dye, when the sample surface was positively charged. The PZC of the unfired Akahoya was approximately 7.0 pH, and it was almost neutral. Therefore, these results suggest that the types of ions adsorbed through the ion exchange function of the natural Akahoya soil change depending on whether the water surrounding the soil is acidified or alkalinized. Although many soils have excellent cation exchange performance, only a few have excellent anion exchange performance. Unfired Akahoya can adsorb both anions and cations in aqueous solutions by adjusting the aqueous solutions’ pH, suggesting that Akahoya can be used to eliminate numerous harmful chemicals from water. 

Second, we discuss the primary factor of Akahoya’s phosphoric acid adsorption. In the phosphoric acid adsorption tests, Akahoya adsorbed some phosphorus in the aqueous solution but clay did not. The measurement results of Akahoya’s zeta potential and pH of the phosphoric acid aqueous solution containing the sample (see Figure 12 and Figure 16(b)) show that the surface of the Akahoya sample in this experiment was negatively charged, confirming that the reaction of Akahoya with phosphoric acid was not via anion exchange. In addition, the Al ion concentration was much higher in the immersion liquid with Akahoya than in the immersion liquid with clay. Therefore, the reaction of Akahoya with phosphoric acid is thought to be mostly due to active Al, as previously explained in many studies on the phosphoric acid adsorption of allophane [9,10,11,12].

The filtration test of *E. coli* and the desorption test after it indicate that Akahoya has the ability to adhere to or hold *E. coli*. The results support the hypothesis that Akahoya can reduce the runoff of *E. coli* from cattle grazing areas to surrounding waters. However, the adhesive strength between Akahoya and *E. coli* is unknown. In addition, this study could not determine whether or not Akahoya has the ability to adsorb *E. coli* in a noncontact state. These are the issues to be clarified in future. 

Akahoya has the potential to be developed into a variety of applied technologies. For example, the processing technology of cartridges using Akahoya will enable environmental purification. Currently, we are also working on developing new antibiotic-free feed additives for the practical application of technology to improve productivity by preventing infectious diseases in livestock and ultimately improving their health.

## 4. Conclusions

To clarify the adsorption ability of Akahoya, the basic chemical and physical properties were measured, and adsorption tests were conducted. The results were used to evaluate the cation/anion exchange performance, gas adsorption performance, and phosphoric acid adsorption performance. Tests were then conducted to evaluate the adhesion and desorption of *E. coli* with Akahoya. The results can be summarized as follows:

Akahoya had lower SiO_2_ and higher Al_2_O_3_ and Fe_2_O_3_ contents than clay that was collected in the same region. The molar ratio of SiO_2_ and Al_2_O_3_ was about 2.7:1. The Akahoya immersion liquid was neutral or weakly alkaline and contained a much higher concentration of Al ions than the clay immersion liquid. These measurements confirmed that Akahoya was not exactly the same soil as Imogo, owing to the higher SiO_2_:Al_2_O_3_ molar ratio and significant amount of Fe. Akahoya was also not exactly the same soil as allophane.

SEM observations indicated that Akahoya mainly contained imogolite and small amounts of a very fine granular mineral similar to allophane. Akahoya contained 50% or more amorphous minerals. Therefore, Akahoya is a mixture of volcanic soil containing imogolite, allophane, and other amorphous minerals with a relatively high Fe content as well as quartz and albite, which are crystalline minerals.

Akahoya possesses a high porosity and an extremely high specific surface area. Fired Akahoya samples had a relatively high specific surface area up to a firing temperature of about 1023 K.

The PZC of the unfired Akahoya was approximately pH 7.0. The PZC of the sample fired at 1073 K was pH 2.9. In other words, the surface of the unfired sample was positively charged when the aqueous solution was acidic and was negatively charged when the solution was alkaline. The sample surface fired at 1073 K was negatively charged except when the aqueous solution was a strong acid, and the sample surfaces fired at above 1273 K were negatively charged at all pH values. The surface potential of Akahoya had a significant effect on the adsorption of acidic and basic dyes.

Akahoya fired at 1073 K had a negatively charged sample surface and, thus, had a high adsorption capacity for MB, which is a basic dye. In contrast, it did not adsorb Orange II and Congo red at all, which are acidic and azo dyes, respectively. When the pH of the aqueous solution was adjusted below the PZC, Akahoya adsorbed Orange II well. These results confirmed that Akahoya had a very high cation exchange capacity when the sample surface was negatively charged and a high anion exchange capacity when the sample surface was positively charged.

Akahoya exhibited a high adsorption capacity for both NO_2_ and SO_2_, which was attributed to the very high specific surface area.

Akahoya exhibited relatively high adsorption capacity for phosphoric acid. The primary factor for the phosphoric acid adsorption of Akahoya is believed to be active Al because its immersion liquid had a very high Al ion concentration. 

The filtration tests of *E. coli* suspension on Akahoya and Shirasu verified that *E. coli* adhered to the Akahoya sample. Furthermore, the *E. coli* release test after the filtration test suggested that the adhesion of *E. coli* to Akahoya could be phosphorus-mediated.

These results indicate that Akahoya can be expected to be used as an adsorbent for fine particles in agricultural and industrial applications.

## Figures and Tables

**Figure 2 nanomaterials-13-00568-f002:**
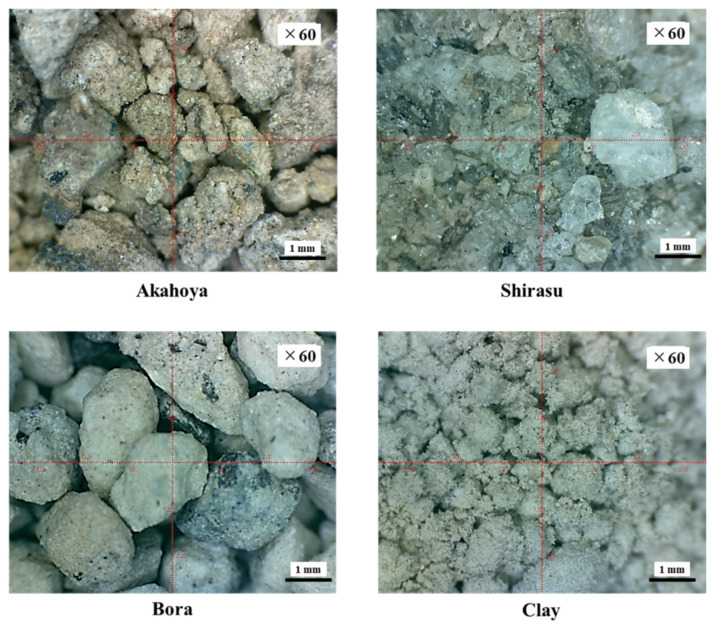
Microscopy images of samples.

**Figure 5 nanomaterials-13-00568-f005:**
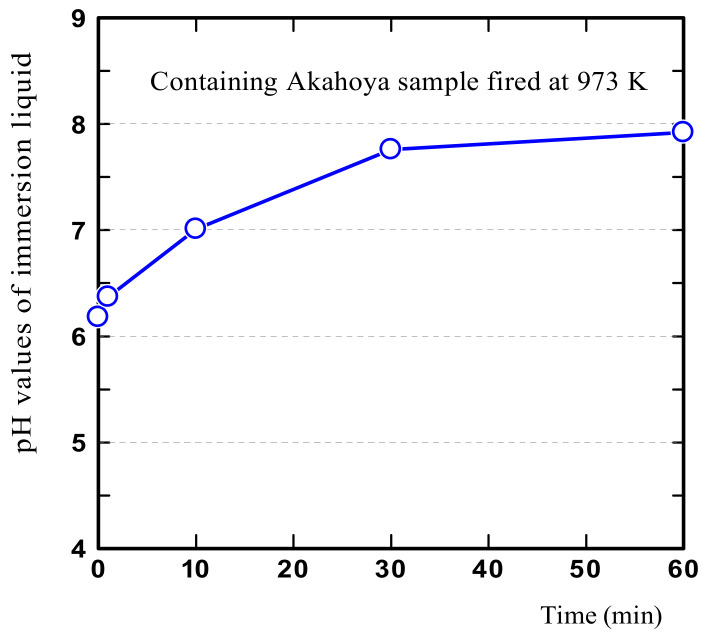
Liquid pH after Akahoya sample immersion.

**Figure 6 nanomaterials-13-00568-f006:**
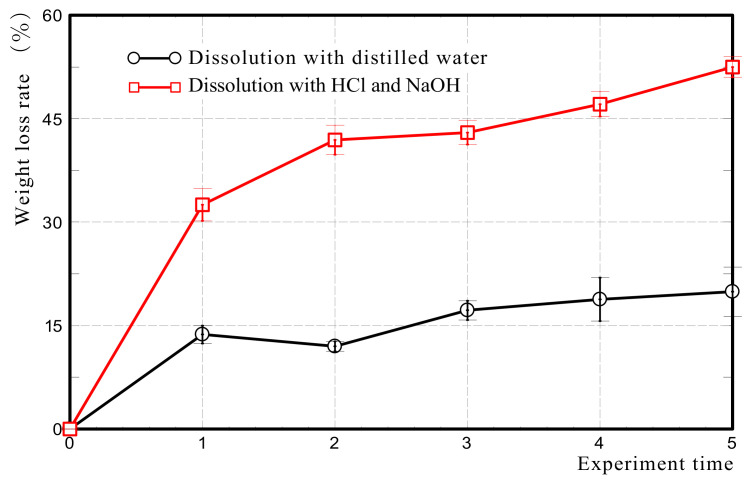
Weight loss rate of Akahoya by the acid–alkali alternate dissolution method.

**Figure 7 nanomaterials-13-00568-f007:**
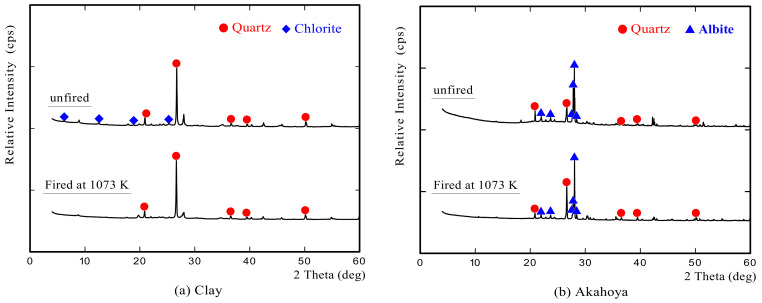
X-ray diffraction profiles of (**a**) clay and (**b**) Akahoya.

**Figure 8 nanomaterials-13-00568-f008:**
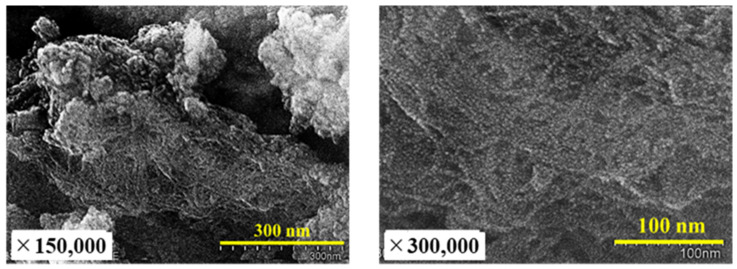
SEM images of the surface structure of Akahoya.

**Figure 9 nanomaterials-13-00568-f009:**
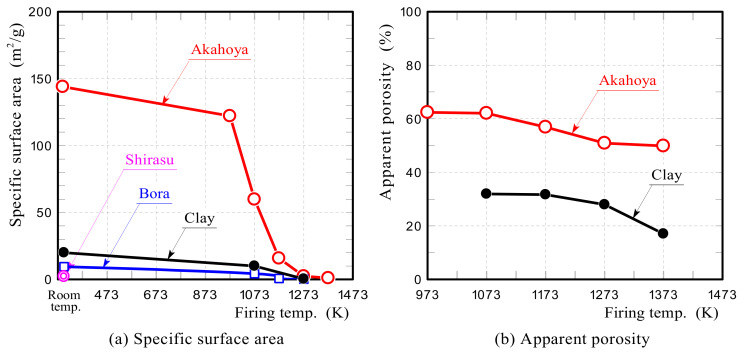
(**a**) Specific surface area and (**b**) apparent porosity of samples.

**Figure 10 nanomaterials-13-00568-f010:**
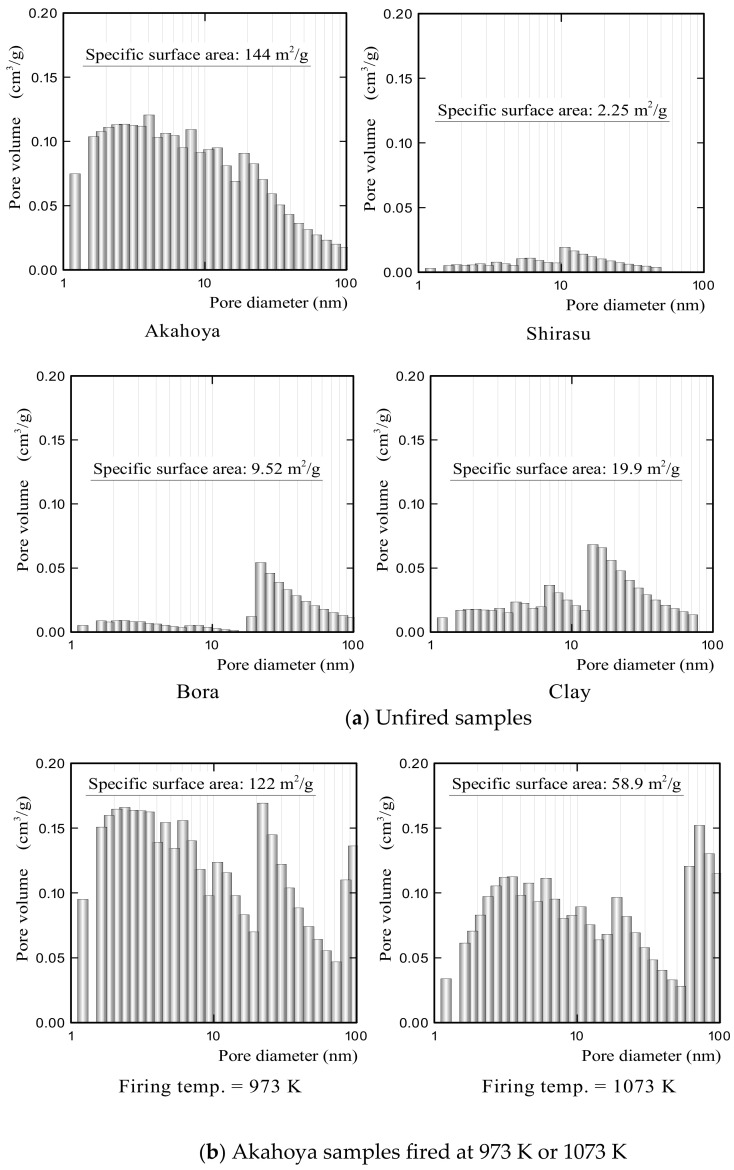
(**a**) Pore size distributions of unfired Akahoya, Shirasu, Bora, and clay samples; (**b**) pore size distributions of Akahoya samples after firing at 973 and 1073 K.

**Figure 11 nanomaterials-13-00568-f011:**
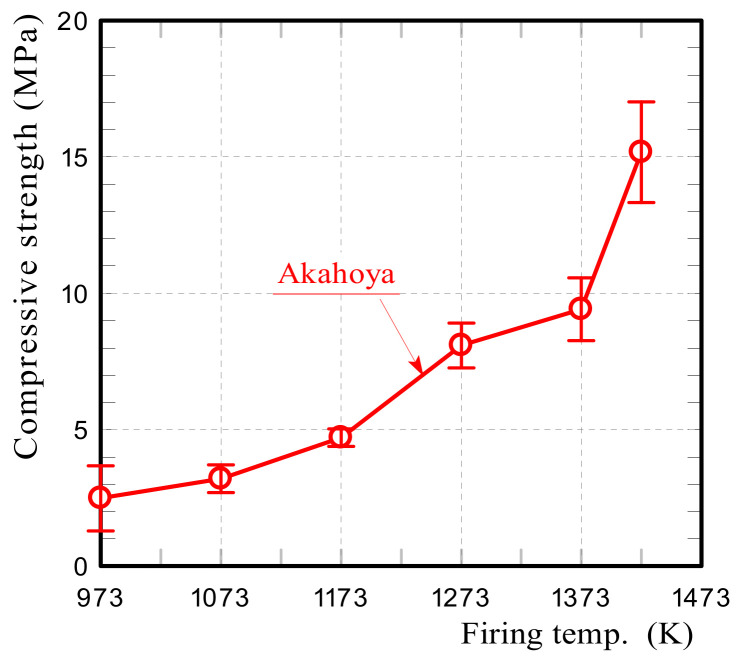
Compressive strength of Akahoya samples fired at 973–1423 K.

**Figure 12 nanomaterials-13-00568-f012:**
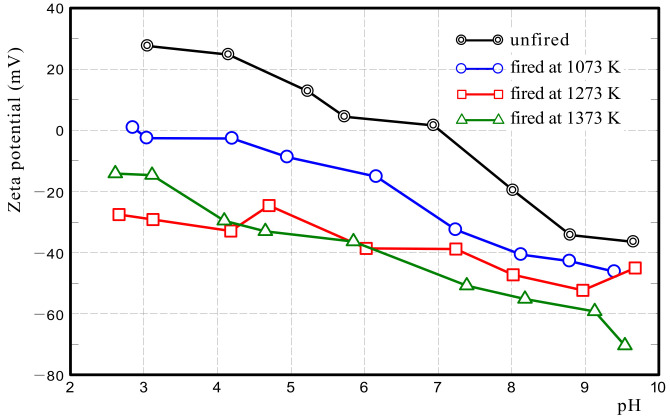
Zeta potential of Akahoya.

**Figure 13 nanomaterials-13-00568-f013:**
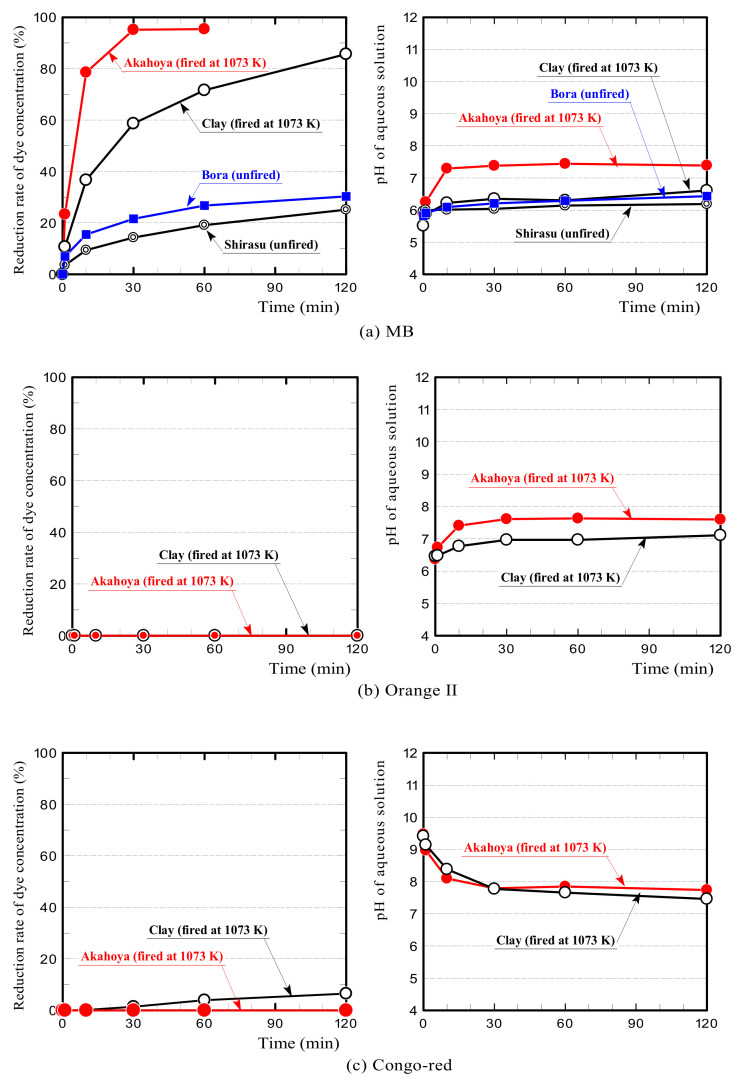
Dye concentration reduction rates of samples and temporal change in pH of the dye solution: (**a**) MB, (**b**) Orange II, and (**c**) Congo red.

**Figure 14 nanomaterials-13-00568-f014:**
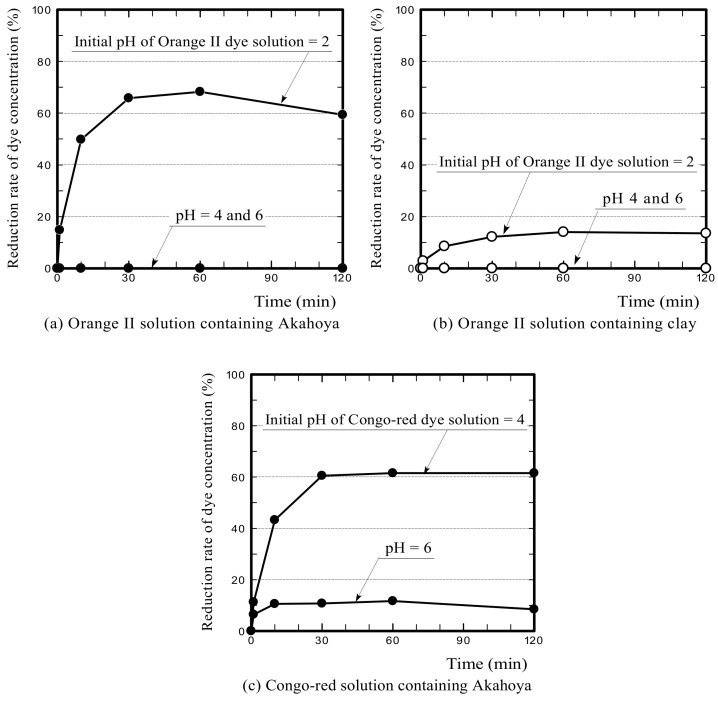
(**a**,**b**) Orange II concentration reduction rates at an initial pH of 2–6 for the dye solution containing the Akahoya or clay sample, respectively; (**c**) reduction rate of the Congo red concentration at an initial pH of 4 or 6 for the dye solution containing Akahoya.

**Figure 15 nanomaterials-13-00568-f015:**
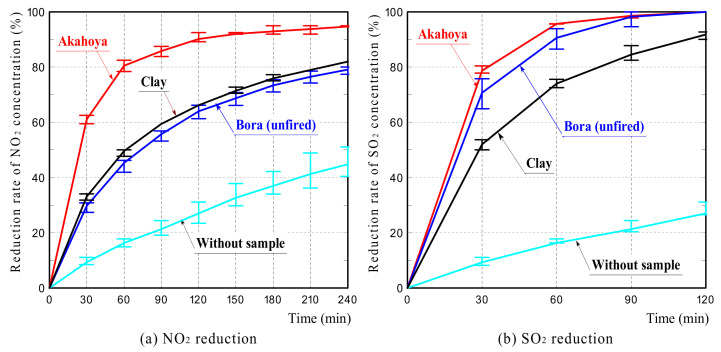
Reduction rates of the (**a**) NO_2_ and (**b**) SO_2_ concentrations with the samples.

**Figure 16 nanomaterials-13-00568-f016:**
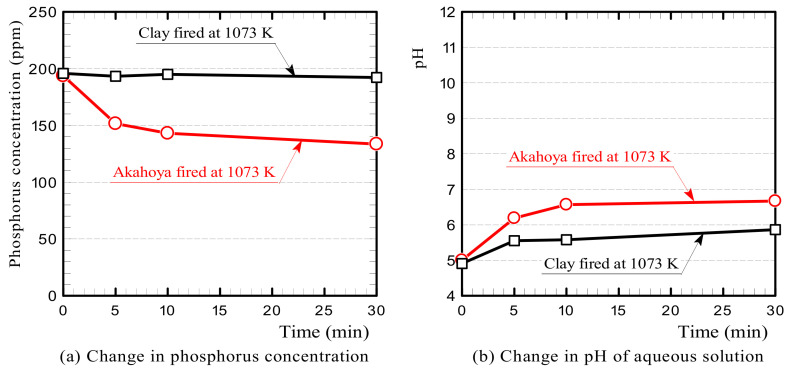
Changes in the (**a**) phosphorus concentration and (**b**) pH of the solution containing the Akahoya or clay sample.

**Figure 17 nanomaterials-13-00568-f017:**
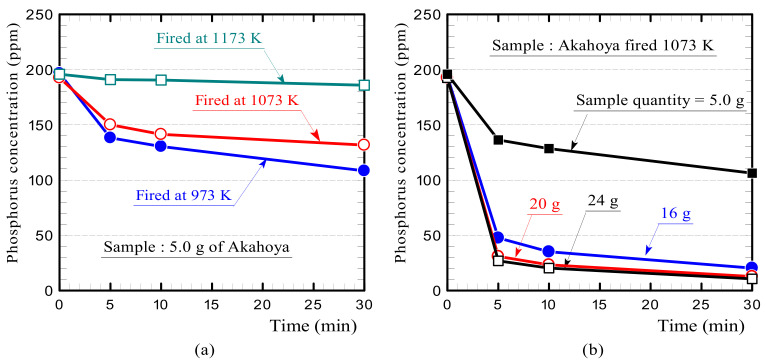
Changes in the phosphorus concentration of the aqueous solution containing Akahoya samples according to the (**a**) firing temperature and (**b**) sample amount.

**Figure 18 nanomaterials-13-00568-f018:**
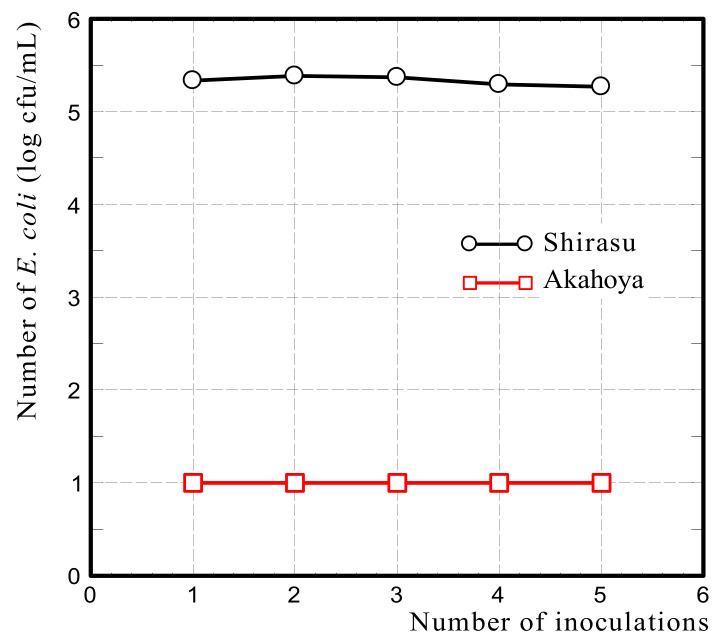
Number of *E. coli* (log cfu/mL) after passing through the column containing Akahoya or Shirasu.

**Figure 19 nanomaterials-13-00568-f019:**
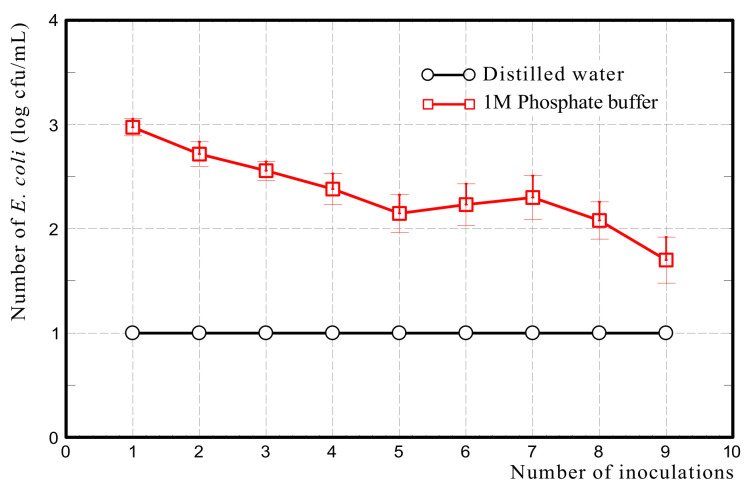
Number of *E. coli* (log cfu/mL) released from Akahoya after the addition of 1 M phosphate buffer or distilled water.

**Table 1 nanomaterials-13-00568-t001:** Inorganic chemical compositions of the volcanic soils and clay.

Type of Soil	Components (Mass%)							
	SiO_2_	Al_2_O_3_	Fe_2_O_3_	K_2_O	MgO	CaO	TiO_2_	Others
Akahoya	50.1	31.5	9.85	1.16	2.97	2.61	1.13	0.67
Shirasu	64.4	27.1	3.63	2.07	-	2.16	0.41	0.7
Bora	67.2	20.1	5.0	2.98	0.77	3.19	0.55	0.18
Clay	65.8	21.9	4.79	3.37	1.67	1.31	0.87	0.29

**Table 2 nanomaterials-13-00568-t002:** Al and Fe ion concentrations in the immersion liquid containing an Akahoya or clay sample.

Immersed Sample	Ion Concentration (ppm)	
	Al	Fe
None (distilled water)	0.014	0.089
Akahoya	0.993	0.422
Clay	0.057	0.115

## Data Availability

The datasets generated and/or analyzed during the current study are available from the first author or corresponding author on reasonable request.

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
