# Peer review of "Fine Particle Adsorption Capacity of Volcanic Soil from Southern Kyushu, Japan"

_nanomaterials, 2023, doi:10.3390/nano13030568_

Round 1

Reviewer 1 Report

The manuscript Fine particle adsorption capacity of volcanic soil from southern Kyushu, Japan presents investigations on Akahoya volcanic soil to assess its main properties to be used as absorbents in agriculture and industry fields. Generally, the manuscript is well written and the investigations were conducted in well-organized work.

As a potential improvement of presentation, I suggest to merge Result and Discussion sections in a single section Results and discussion. The results were generally well discussed in the Results section, and the Discussion section remained too short in my opinion.

Conclusions section: This can be presented without numbering the paragraphs. Also, a main conclusion with some perspectives in usage of these results may be interesting.

Author Response

Dear reviewer,

Subject: Submission of revised paper “nanomaterials-2172489”

We wish to express our strong appreciation to you for careful reading our manuscript and for giving useful comments. We have carefully reviewed the comments and have revised the manuscript accordingly. The following is a point-by-point response to the comments.

Comments

The manuscript Fine particle adsorption capacity of volcanic soil from southern Kyushu, Japan presents investigations on Akahoya volcanic soil to assess its main properties to be used as absorbents in agriculture and industry fields. Generally, the manuscript is well written and the investigations were conducted in well-organized work.

As a potential improvement of presentation, I suggest to merge Result and Discussion sections in a single section Results and discussion. The results were generally well discussed in the Results section, and the Discussion section remained too short in my opinion. Conclusions section: This can be presented without numbering the paragraphs. Also, a main conclusion with some perspectives in usage of these results may be interesting.

Response

In response to your suggestion, we have merged "Results" and "Discussion" into "Results and Discussion. Also, we have removed the numbering in the Conclusions section. In addition, the following sentence was added at the end.

P.19, line 616; “These results indicate that Akahoya can be expected to be used as an adsorbent for fine particles in agricultural and industrial applications.”

Thank you once again for your valuable comments and suggestions. We are grateful for the time and energy you expended on our behalf.

Sincerely yours,

Hiroyuki Kinoshita

University of Miyazaki Japan

Reviewer 2 Report

The paper Fine particle adsorption capacity of volcanic soil from southern Kyushu, Japan refers to the use of this soil for adsorption of various fine particles.

Abstract is clear and concise. Introduction is well written in which authors explain what was the reason for using this soil as adsorbent. Experimental part presents each step of research.  A lot of experiment were done and the obtained results are conclusive. Discussion are short but clear. Conclusions are sound. But, before acceptance some clarifications are needed.

In the paper you specified that:”Akahoya" is a volcanic soil classified as a special soil deposited in Kyushu, Japan having high adsorption capacity for diverse chemicals, gases and  Escherichia coli .”

But in fact in almost all experiment you used : fired powdered Akahoya to produce granular ceramic samples. 

 Please clarify this aspect.

 Oxidation firing term is confusing.

Author Response

Dear reviewer,

Subject: Submission of revised paper “nanomaterials-2172489”

We wish to express our strong appreciation to you for careful reading our manuscript and for giving useful comments. We have carefully reviewed the comments and have revised the manuscript accordingly. The following is a point-by-point response to the comments.

Comments

The paper Fine particle adsorption capacity of volcanic soil from southern Kyushu, Japan refers to the use of this soil for adsorption of various fine particles. The abstract is clear and concise. The introduction gives a good explanation of the reasons for using this soil as an adsorbent. In the experimental section, each step of the research is presented. Many experiments were conducted and the results obtained are conclusive. The discussion is brief but clear. The conclusions are sound. However, some clarification is needed before acceptance. The paper clearly states that "Akahoya is a special volcanic soil deposited in Kyushu that has a high adsorption capacity for a wide variety of chemicals, gases, and E. coli.

In reality, however, most experiments use calcined powdered akahoya to prepare granular ceramic samples. Please clarify this point. The term "oxidative calcination" is confusing.

Response

When using Akahoya as an adsorbent, it is desirable to use it without calcination because calcination lowers the specific surface area. However, problems may arise when powder form samples are used as adsorbent. For example, the immersion liquid of a powder form sample is turbid, so filtration of the turbid liquid after adsorption of fine particles is necessary. This filtration takes time. In gas adsorption, the amount of gas that can pass between samples is smaller. A separate dust collector may also be required because the powder disperses into the gas. We believe that if powder form samples could be formed so that they could be easily used as adsorbent, the use in agriculture and industry would increase. For these reasons, we used samples of Akahoya ceramics in this experiment. The reason for the preparation of granular ceramics is already described on page 3, line 112.

In Fig. 1, "oxidative" has been removed. Special volcanic soil" was revised to "Akahoya or clay soil".

Thank you once again for your valuable comments and suggestions. We are grateful for the time and energy you expended on our behalf.

Sincerely yours,

Hiroyuki Kinoshita

University of Miyazaki Japan

Reviewer 3 Report

1. Please add more numerical data to the abstract section

2. Fig.2 please add the magnification to the figure

3. Fig.8 please enhance the resolution of used Sem microphotographs. Additionally, add magnification and if possible EDS mapping 

4. Apart from showing the graphs of the surface area please provide the numerical value of the maximum specific surface area in the text

5. Fig.14 a) Why after 60 minutes the desorption phenomena occur?

6. Did the authors try any of the equilibrium isotherm modeling and kinetic modeling of adsorption processes?

Author Response

Dear reviewer,

Subject: Submission of revised paper “nanomaterials-2172489”

Please refer to the attached response form.

Thank you once again for your valuable comments and suggestions. We are grateful for the time and energy you expended on our behalf.

Sincerely yours,

Hiroyuki Kinoshita

University of Miyazaki Japan
